# Electrochemically coupled CH$_4$ and CO$_2$ consumption driven by microbial processes

Yue Zheng[1,2,9], Huan Wang [1,2,9], Yan Liu[3,4], Peiyu Liu[3,4], Baoli Zhu[5], Yanning Zheng[6], Jinhua Li [3,4], Ludmila Chistoserdova[7], Zhiyong Jason Ren [8] ✉ & Feng Zhao [1] ✉

The chemical transformations of methane (CH$_4$) and carbon dioxide (CO$_2$) greenhouse gases typically have high energy barriers. Here we present an approach of strategic coupling of CH$_4$ oxidation and CO$_2$ reduction in a switched microbial process governed by redox cycling of iron minerals under temperate conditions. The presence of iron minerals leads to an obvious enhancement of carbon fixation, with the minerals acting as the electron acceptor for CH$_4$ oxidation and the electron donor for CO$_2$ reduction, facilitated by changes in the mineral structure. The electron flow between the two functionally active microbial consortia is tracked through electrochemistry, and the energy metabolism in these consortia is predicted at the genetic level. This study offers a promising strategy for the removal of CH$_4$ and CO$_2$ in the natural environment and proposes an engineering technique for the utilization of major greenhouse gases.

Carbon dioxide (CO$_2$) and methane (CH$_4$) are the two dominant greenhouse gases, collectively accounting for over 90% of the total radiative forcing from all greenhouse gases[1]. The valence of carbon in CH$_4$ and CO$_2$ are −4 and +4, respectively, which represents the lowest and highest valences of carbon. The energy of the C−H bond in CH$_4$ is 413 kJ/mol, while the energy of the C=O bond in CO$_2$ is 799 kJ/mol[2], indicating that significant energy input is necessary for the fixation of both gases[3,4]. For instance, catalytic reactions involving CH$_4$ oxidation and CO$_2$ reduction typically require high temperatures exceeding 700 °C and often result in relatively low yields[5]. Therefore, breaking the energy barrier to simultaneously conduct CH$_4$ oxidation and CO$_2$ reduction at normal pressure and room temperature has been reported as difficult.

Despite being invisible to the naked eye, microbes are abundant (~10$^{30}$ cells) and microbial processes are fundamental to the functioning of ecosystems, driving key processes such as element cycling, nutrient decomposition, and organic matter transformation[6]. In contrast to extreme physicochemical methods, the biological fixation of these gases is largely dependent on the natural ability of microorganisms using highly specialized enzymes. The primary microbial sink for CH$_4$ is a specialized group of microbes known as the methanotrophs[7], which play a crucial role in regulating the atmospheric CH$_4$ budget[8]. The methanotrophs potentially couple CH$_4$ oxidation to multiple electron acceptors including O$_2$, Fe$^{3+}$ or Mn$^{4+}$ [9–12]. On the other hand, CO$_2$-reducing bacteria have been isolated since the 1930s[13], and there are six known biochemical pathways for carbon fixation[14]. Microbes such as *Rhodopseudomonas* can reduce CO$_2$ by extracting electrons from minerals, such as Fe$^{2+}$ [15]. Therefore, being able to regulate microbial processes is considered as a potential solution for mitigating greenhouse gas emissions in nature and man-made systems[16,17].

[1]CAS Key Laboratory of Urban Pollutant Conversion, Institute of Urban Environment, Chinese Academy of Sciences, Xiamen 361021, China. [2]State Key Laboratory of Marine Environmental Science, and College of the Environment and Ecology, Xiamen University, Xiamen 361102, China. [3]Key Laboratory of Earth and Planetary Physics, Institute of Geology and Geophysics, Chinese Academy of Sciences, Beijing 100029, China. [4]Laboratory for Marine Geology, Qingdao Marine Science and Technology Center, Qingdao 266237, China. [5]Key Laboratory of Agro-ecological Processes in Subtropical Regions and Taoyuan Agro-ecosystem Research Station, Institute of Subtropical Agriculture, Chinese Academy of Sciences, Changsha 410125, China. [6]State Key Laboratory of Microbial Resources, Institute of Microbiology, Chinese Academy of Sciences, Beijing 100101, China. [7]Department of Chemical Engineering, University of Washington, Seattle, WA, USA. [8]Department of Civil and Environmental Engineering, and Andlinger Center for Energy and the Environment, Princeton University, 41 Olden St., Princeton, NJ 08540, USA. [9]These authors contributed equally: Yue Zheng, Huan Wang. ✉e-mail: zjren@princeton.edu; fzhao@iue.ac.cn

From the perspective of electron transport, the $CH_4$ sink is achieved through the oxidation process by taking electrons from methane[18], the $CO_2$ sink is achieved through the reduction reaction with carbon dioxide accepting electrons from electron donor[19]. The collaboration between microorganisms weaves into a complex network of interactions that collectively drive the biogeochemical cycle of carbon, e.g. interspecies extracellular electron transfer is known as a classical microbial interaction, and is an important way of coupling two opposite electron transfer processes take place in the same system. However, it remains unknown whether $CH_4$- and $CO_2$-metabolizing microbial consortia could collaborate via interspecies extracellular electron transfer at temperate conditions.

In this study, we investigate the feasibility and the potential role of iron in linking $CH_4$ oxidation and $CO_2$ reduction. We employ both microcosm and enrichment cultures to monitor carbon fixation as well as the redox cycle of iron minerals. The changes in mineral composition are followed by characterizing magnetic properties and the lattice structure of iron minerals. Based on the changes observed, we propose that iron minerals can act as an energy bridge between the $CH_4$-oxidizing and the $CO_2$-reducing consortia. In addition, we construct bioelectrochemical systems to track the electron transport by combining $CH_4$ oxidation in the anode and $CO_2$ reduction in the cathode. Our results demonstrate that the co-metabolism of $CH_4$ and $CO_2$ may occur within one system at temperate conditions, providing a promising strategy for regulating sinks of these major greenhouse gases by engineering design.

## Results

### $CH_4$ oxidation and $CO_2$ reduction promoted by iron minerals

We hypothesized that the processes of $CH_4$ oxidation and $CO_2$ reduction may be linked, through the regenerative action of iron minerals. We first tested this hypothesis by employing paddy soil samples on the water-soil interface. The dark/light conditions were used as a switch from the $CH_4$ oxidizing (dark) to $CO_2$ reducing (light) activity of the natural soil community. In the dark, ferrihydrite has been reported to serve as an alternative electron acceptor after

dioxygen became low to drive $CH_4$ oxidation, and in an additional experiment we also observed such switch (Supplementary Fig. 12)[20]. In our experimental setup, $46.32 \pm 6.05\%$ of $CH_4$ was consumed in the presence of ferrihydrite, versus $25.53 \pm 5.14\%$ of that without ferrihydrite (Fig. 1a). Under light, $57.45 \pm 6.49\%$ of $CO_2$ was consumed in the presence of ferrihydrite, while only $31.04 \pm 5.68\%$ was consumed in the absence of ferrihydrite (Fig. 1a). Compared with the killed control group, $16.21 \pm 1.89\%$ of Fe(III) was reduced to Fe(II) after $CH_4$ oxidation, while the proportion of Fe(II) returned to $2.14 \pm 0.54\%$ after $CO_2$ reduction (Supplementary Fig. 1). For the electron flow, $18.94 \pm 9.22\%$ of electron in $CH_4$ was used to iron reduction, $28.81 \pm 8.52\%$ of electron in $CO_2$ was used to iron oxidation, and the remains were to produce biomass and other metabolites. Microbial aggregates were observed on the surface of the iron mineral by scanning electron microscopy (Supplementary Fig. 2a). The aggregates facilitate the redox reactions of iron minerals via cell-mineral interactions, subsequently enabling the oxidation of $CH_4$ or the reduction of $CO_2$. In the absence of ferrihydrite, the $CH_4$ and $CO_2$ were also consumed via the mediation organic electron mediators, though on a much less scale. This was supported by the changes of total organic carbon content (Supplementary Fig. 2b). These results indicated that the consumption of both $CH_4$ and $CO_2$ was promoted by the iron mineral which may be serving as a natural geo-battery, acting as an electron source and electron sink, connecting two redox processes to couple the $CH_4$ oxidation and the $CO_2$ reduction (Fig. 1b), akin to a previously described process of magnetite reduction/oxidation by different species of iron-metabolizing bacteria[21].

### The redox cycle of iron minerals during co-metabolism of $CH_4$ and $CO_2$

Considering the complexity of microcosm systems, the role of iron minerals in the respective redox cycles was investigated using enriched microbial consortia, the ones active in $CH_4$ oxidation and the other active in $CO_2$ reduction. Under dark, $CH_4$ is used as an electron donor and ferrihydrite as an electron acceptor. After 8 days, the ratio of Fe(II)/Fe(total) increased from $1.90 \pm 0.01\%$ (Fe$_{\text{initial}}$) to $82.20 \pm 2.22\%$

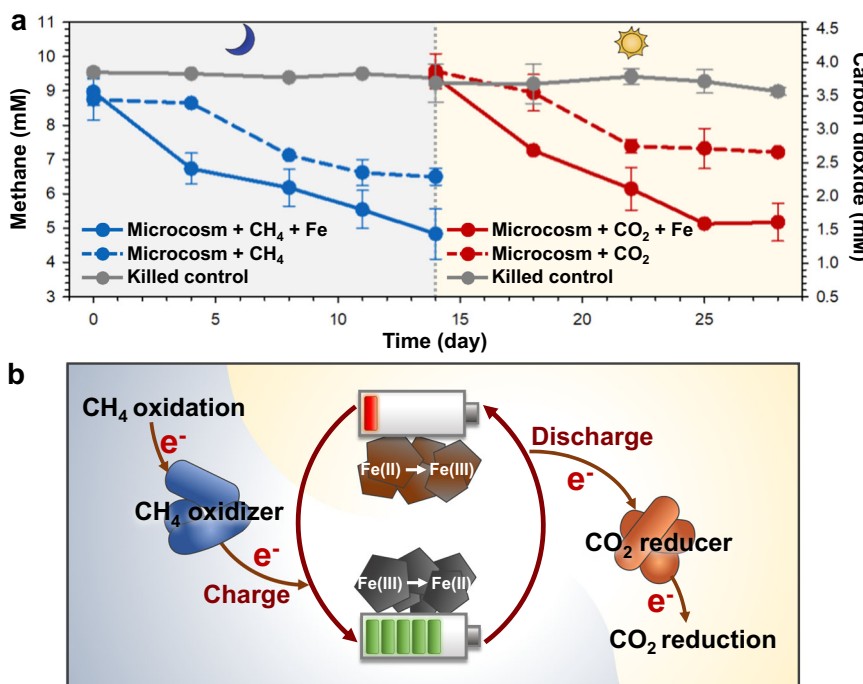

**Fig. 1 | $CH_4$ oxidation and $CO_2$ reduction in microcosms, promoted by iron minerals. a** $CH_4$ oxidation and $CO_2$ reduction with and without iron minerals in microcosm incubations. The killed control represents the incubation with sterilized soil. Data generated from $n = 3$ biologically independent samples for each group and error bars indicate standard deviation of the mean. **b** Proposed mechanism for iron minerals serving as a geo-battery to couple $CH_4$ oxidation and $CO_2$ reduction.

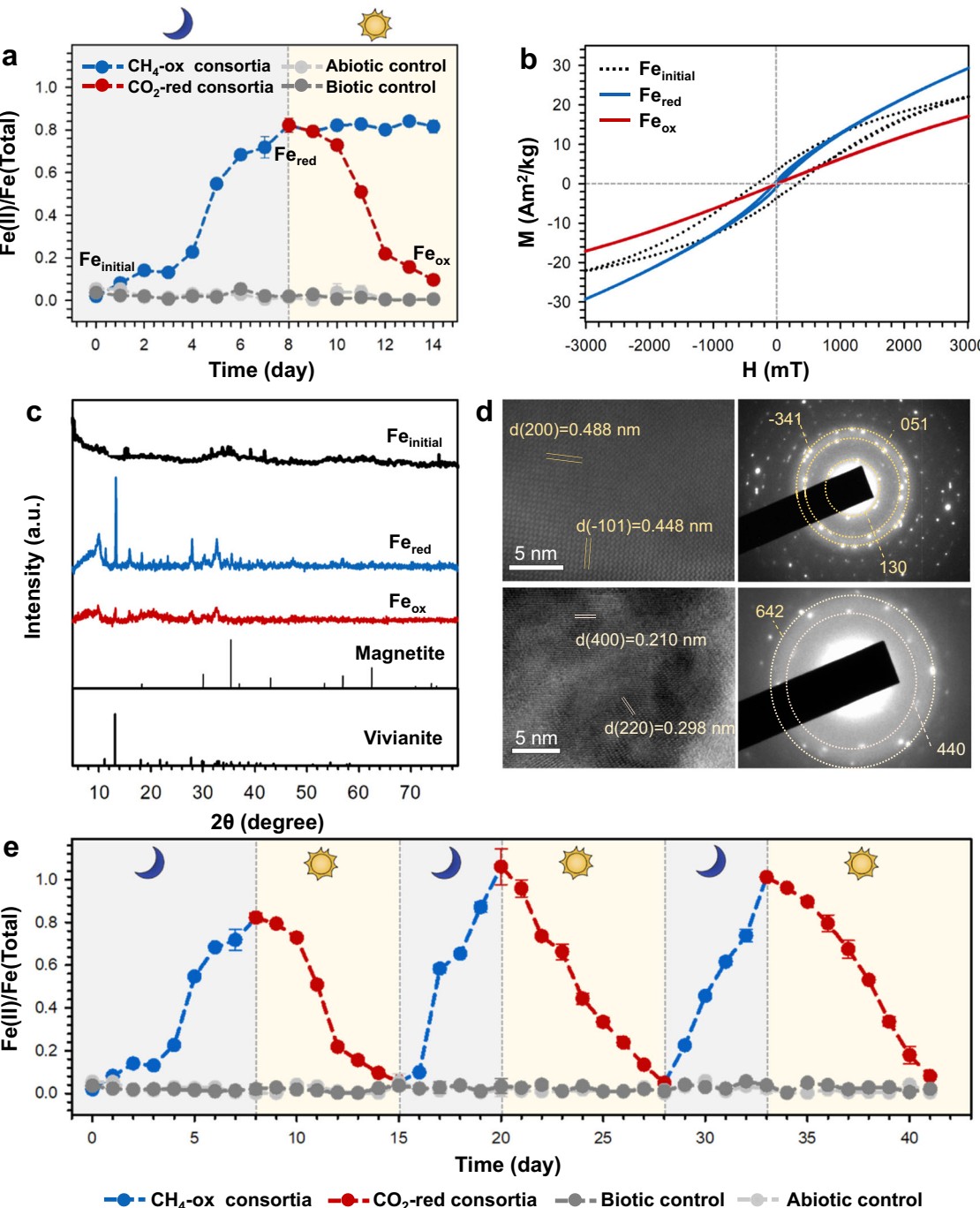

**Fig. 2 | The redox cycles of iron minerals driven by CH₄/CO₂-metabolizing consortia.** **a** Observed changes in Fe(II)/Fe(total) over time in iron minerals. The redox cycle of iron minerals was switched by light/dark conditions. The blue line means the participation of CH₄-oxidizing consortia (CH₄-ox consortia), and the red line means the participation of CO₂-reducing consortia (CO₂-red consortia). The abiotic control represents the incubation without cells, and the biotic control represents the incubation without CH₄ and CO₂. Fe$_{initial}$, Fe$_{red}$, and Fe$_{ox}$ represent the initial ferrihydrite, reduced iron minerals, and oxidized iron minerals. Data generated from $n = 3$ biologically independent samples for each group and error bars indicate standard deviation of the mean. **b** Magnetic hysteresis curves of three samples from Fe$_{initial}$, Fe$_{red}$, and Fe$_{ox}$. **c** X-ray diffraction of three samples from Fe$_{initial}$, Fe$_{red}$, and Fe$_{ox}$. **d** Selected area electron diffraction and lattice images of samples collected at Fe$_{red}$. The images of Fe$_{intial}$ and Fe$_{ox}$ were listed in the supporting information. **e** Observed changes of Fe(II)/Fe(total) driven by CH₄-oxidizing consortia (blue line, CH₄-ox consortia), under dark, and by CO₂-reducing consortia (red line, CO₂-red consortia), under light. The abiotic control represents the incubation without cells, and the biotic control represents the incubation without CH₄ and CO₂. Data generated from $n = 3$ biologically independent samples for each group and error bars indicate standard deviation of the mean.

(Fe$_{red}$) (Fig. 2a). There were $5.88 \pm 0.21$ mM of CH₄ consumed, and the amount of electrons that were transferred from CH₄ to ferrihydrite was $17.08 \pm 0.48\%$ (Supplementary Fig. 3a). According to magnetic hysteresis loops (Fig. 2b), magnetic ability of the iron mineral was enhanced after it was reduced by CH₄-oxidizing consortia, based on the increase in saturation magnetization (Ms) of 7.31 Am²/kg between Fe$_{initial}$ and Fe$_{red}$. It appears that a paramagnetic mineral may be formed, as the saturation remanence (Mrs) and near-zero coercivity (Hc) were decreased to 0.75 Am²/kg and 40.21 mT, respectively (Supplementary Table 1). Based on the X-ray diffraction data, the lattice of

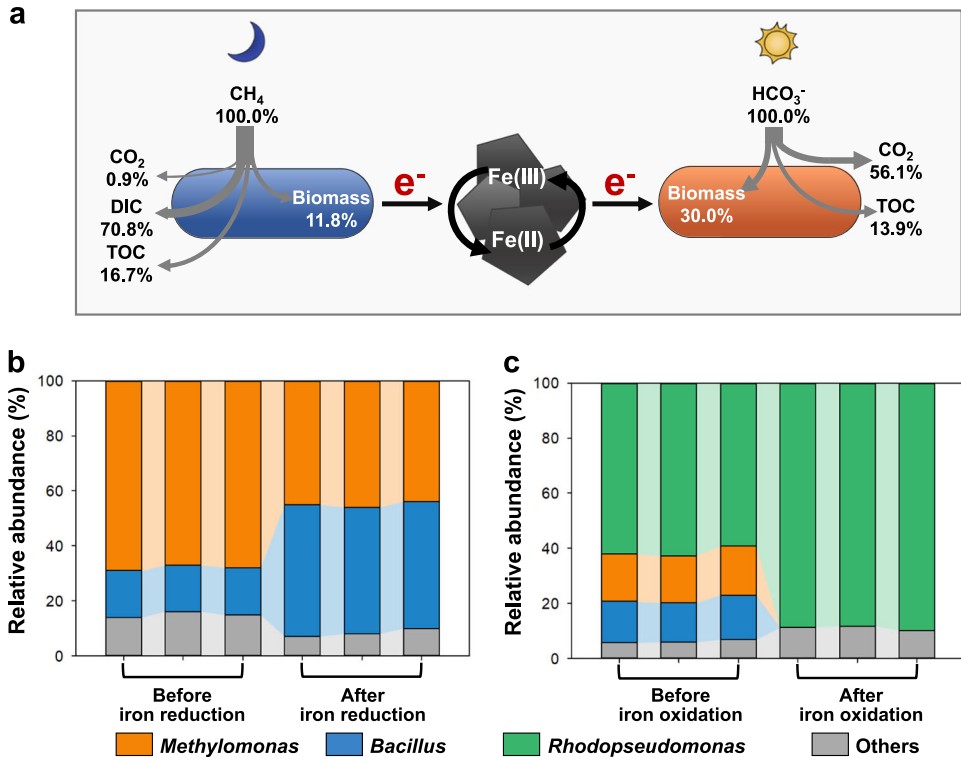

**Fig. 3 | The carbon flow and microbial composition in CH₄/CO₂-metabolizing consortia. a** The carbon flow of the proposed process, in which CH₄ and bicarbonate were converted into different carbon fractions. **b** The composition of CH₄-oxidizing consortia before and after iron reduction. **c** The composition of CO₂-reducing consortia before and after iron oxidation. Data generated from $n = 3$ biologically independent samples for each group.

vivianite and magnetite appeared to be at Fe$_{red}$, along with the (110), (020) and (031) of vivianite, (111), (220) and (311) of magnetite (Fig. 2c). The lattice information was further confirmed by high-resolution transmission electron microscopy, indicating that vivianite and magnetite co-existed in the reduced minerals in conjunction with the oxidation of CH₄ (Fig. 2d).

When the incubations were shifted to the light condition, CO₂-reducing consortia participated in the subsequent process, using oxidized iron minerals as electron donors and bicarbonate as an electron acceptor. The Fe(II)/Fe(total) decreased to $9.56 \pm 0.23\%$ (Fe$_{ox}$) over 6 days (Fig. 2a), which meant iron minerals returned to the reduced state. At this time, $7.30 \pm 0.80$ mM of CO₂ was consumed, and $26.43 \pm 2.23\%$ of the electrons were transferred from iron minerals to bicarbonate (Supplementary Fig. 3b), while without the participation of CO₂-reducing consortia, the Fe(II)/Fe(total) of minerals maintained the level of Fe$_{red}$ under light condition (Fig. 2a). The magnetic ability was weakened, based on the decrease in $Ms$ at Fe$_{ox}$ of 12.18 Am²/kg, as compared to Fe$_{red}$ (Supplementary Table 1). Part of the iron mineral was still paramagnetic, with the $Mrs$ decrease to 0.03 Am²/kg (Fig. 2b and Supplementary Table 1). The results of XRD showed that the lattices of vivianite and magnetite were weakened at Fe$_{ox}$ than that of Fe$_{red}$ (Fig. 2c), and both were observed by lattice image microscopy and electron diffraction (Fig. 2d and Supplementary Fig. 4). Phosphate was one key component in hydrated vivianite Fe₃(PO₄)₂·8H₂O. In the incubation system without phosphate, the only mineral was magnetite after reduction (Supplementary Fig. 5b–d). The maximum ratio of Fe(II)/Fe(total) was $82.20 \pm 2.22\%$ and $58.20 \pm 0.41\%$ for the culture system with phosphate and without phosphate, respectively (Supplementary Fig. 5a). This suggests that the presence of phosphate may be enhancing iron transformation during the redox cycling.

After the first cycle of iron transformation, the regenerability of iron minerals and the repeatability of the redox cycle were verified by two more cycles (Fig. 2e). During the second and the third cycles, iron

minerals were completely reduced, 100% Fe(II)/Fe(total), by the CH₄-oxidizing consortia under the dark condition. After the switch to the light condition, the iron mineral was oxidized again to $5.51 \pm 0.21\%$ Fe(II)/Fe(total), driven by the CO₂-reducing consortia. In the absence of microbial cells (the abiotic control) or the carbon source (the biotic control), the ratio of Fe(II)/Fe(total) remained constant during the entire cycle. These results demonstrated the reversibility and the sustainability of bidirectional electron transfers, to and from iron mineral, during the redox cycle, further supported by the observed and reversible changes in the lattice structure (Fig. 2d and Supplementary Fig. 4).

## The carbon flow and microbial composition during co-metabolism of CH₄ and CO₂

The carbon flow was monitored during the co-metabolism of CH₄ and CO₂ through isotopic labeling (Fig. 3a). In these experiments, a total of $52.4 \pm 2.5\%$ CH₄ and $41.19 \pm 1.31\%$ bicarbonate was consumed. $16.7 \pm 6.0\%$ of CH₄ was converted to dissolved organic carbon and $11.8 \pm 1.7\%$ to biomass, while $13.9 \pm 0.5\%$ of bicarbonate was converted into dissolved organic carbon, $30.0 \pm 4.8\%$ to biomass, and $56.1 \pm 3.5\%$ to CO₂ due to the pH decreased from $7.87 \pm 0.18$ to $7.25 \pm 0.04$. There was no carbon conversion in the absence of cells (the abiotic control) or the carbon source (the biotic control) during the entire process, indicating that the carbon flow must be produced by microbial activity.

According to the 16S rRNA gene fragment amplicon sequencing, the CH₄-oxidizing consortia consisted of *Methylomonas* ($68.3 \pm 1.4\%$), *Bacillus* ($16.9 \pm 0.5\%$) and other bacteria ($14.8 \pm 1.4\%$). The *Methylomonas* population decreased to $45.0 \pm 1.2\%$, while the *Bacillus* population increased to $46.8 \pm 0.7\%$ after the iron reduction phase (Fig. 3b). *Methylomonas* is one type of the methanotrophic bacteria, and it is frequently found in soils, sediments and wetlands[22]. *Bacillus* species are known to be electrochemically active, being able to use extracellular minerals as electron acceptors[23,24]. The CO₂-reducing consortia

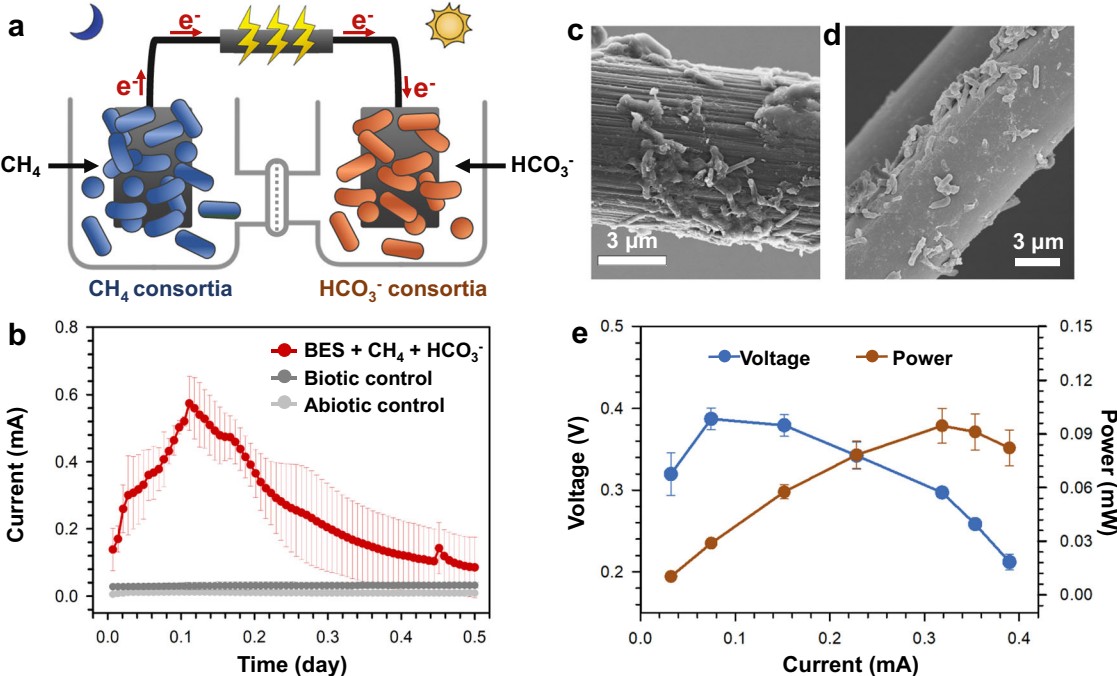

**Fig. 4 | The proposed electron flow between the CH₄-oxidizing and the CO₂-reducing consortia. a** The bioelectrochemical system used for demonstrating the electron flow. **b** Electrical output of the bioelectrochemical system (BES) connecting CH₄-oxidizing and CO₂-reducing consortia, and corresponding controls that is without CH₄/HCO₃⁻ (Biotic control) or without cells (Abiotic control). Data generated from $n = 3$ biologically independent samples for each group and error bars indicate standard deviation of the mean. **c** An image of scanning electron microscope of the biofilm attached on the anode. **d** An image of scanning electron microscope of the biofilm attached on the cathode; representative of 12 images. **e** The polarization curve of this bioelectrochemical system. Data generated from $n = 3$ biologically independent samples for each group and error bars indicate standard deviation of the mean.

were dominated by *Rhodopseudomonas* ($61.3 \pm 1.9\%$), whose population increased to $88.9 \pm 0.8\%$ after iron oxidation (Fig. 3c). *Rhodopseudomonas* species are metabolically versatile and are known to harness light for gaining energy, which is typical Fe(II)-oxidizer with a metabolic pattern of photoferrotrophs[25]. We hypothesized that the CH₄-oxidizing consortia (mainly *Methylomonas* and *Bacillus* in our experiments), were collaborating with other bacteria, using CH₄ as an electron donor and passing some electrons to the extracellular iron minerals, while the CO₂-reducing consortia (mainly *Rhodopseudomonas*) received the electrons from the extracellular iron minerals, and used bicarbonate as an electron acceptor.

## The electron flow during co-metabolism of CH₄ and CO₂

To further understand and track the electron flow between CH₄-oxidizing and the CO₂-reducing consortia, two-chamber bioelectrochemical reactors were constructed and operated (Fig. 4a). CH₄-oxidizing consortia (electron-donating, Fig. 4c) and CO₂-reducing consortia (electron-accepting, Fig. 4d) were attached to the anode and the cathode, respectively, separated by an ion exchange membrane. The anode and the cathode were connected by a conductive wire and external resistance. When fixed by 500 Ω external resistance, the electrical output between the anode and the cathode could reach up to $0.57 \pm 0.08$ mA (Fig. 4b). This is accompanied by anodic CH₄ consumption at $5.42 \pm 0.35$ mM and cathodic bicarbonate consumption at $3.96 \pm 0.43$ mM (Supplementary Fig. 6). The pH of the outflow in both the anodic and the cathodic chambers was stable, suggesting that pH was not playing a role in the electrical output (Supplementary Fig. 7a). The absence of cells (the abiotic control) or the carbon source (the biotic control) in the anodic and the cathodic chambers resulted in no observable electrical output (Fig. 4b). The polarization curve the system shows that with the increase of current, the maximum output voltage reached $0.387 \pm 0.013$ V, and the maximum output power

reached $0.095 \pm 0.010$ mW (Fig. 4e). These results indicated that the electrical output must be generated by the combined action of the CH₄-oxidizing consortia on the anode and the CO₂-reducing consortia on the cathode.

## Electrochemical studies with pure bacterial cultures

To better understand the mechanism of the electron transfer by the CH₄-oxidizing consortia, considering the composition of the anodic biofilm (Fig. 4c), the dominant species, *Methylomonas* and *Bacillus*, were isolated in pure cultures from the CH₄-oxidizing consortia, and these were named *Methylomonas* sp. WH-1 and *Bacillus* sp. WH-2, respectively (Supplementary Fig. 8). The electrical output of *Methylomonas* sp. WH-1 was weak (Fig. 5a), consistent with the previous study[20]. At the same time, we could not detect an electrical signal from *Bacillus* sp. WH-2 when CH₄ was used as a carbon source (Fig. 5a). However, when combined, the co-culture of *Methylomonas* sp. WH-1 and *Bacillus* sp. WH-2 used CH₄ as an electron donor (Supplementary Fig. 10a), and produced an electric current as strong as $0.20 \pm 0.01$ mA (Fig. 5b). The supernatant of *Methylomonas* sp. WH-1 could also drive *Bacillus* sp. WH-2 to reduce ferrihydrite (Supplementary Fig. 9a) and generate electric current (Supplementary Fig. 9b). It has been previously suggested that methanotrophs release organics, specifically volatile fatty acids such as formate, acetate that may benefit other species[26,27]. Thus, we measured the consumption of a select number of fatty acids in the supernatants of *Methylomonas* sp. WH-1 (Fig. 5c). Based on the prominent presence of acetate, we propose that the cross-feeding interaction may be taking place between the two species through acetate, and that *Bacillus* sp. WH-2 generates electrical current when acetate is available (Fig. 5d). The presumed cross-feeding between *Methylomonas* sp. WH-1 and *Bacillus* sp. WH-2 was further corroborated by identifying the respective pathways in each species through genomic annotation (Supplementary Fig. 10b). In addition,

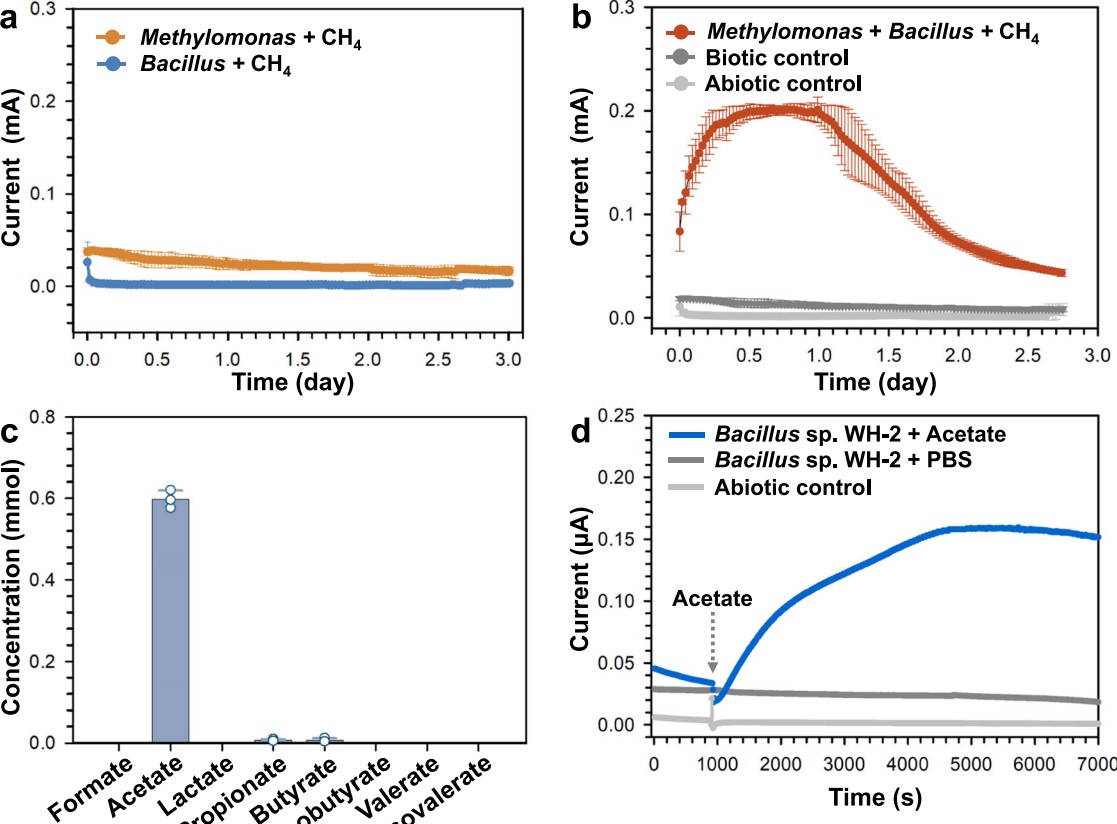

**Fig. 5 | Electron transfer proposed for the interdependent CH₄-oxidizing consortia based on pure culture studies. a** Electrical output of pure cultures of *Methylomonas* sp. WH-1 and *Bacillus* sp. WH-2, individually, in the bioelectrochemical system. Data generated from $n = 3$ biologically independent samples for each group and error bars indicate standard deviation of the mean. **b** Electrical output of the co-culture of *Methylomonas* sp. WH-1 and *Bacillus* sp. WH-2 in the bioelectrochemical systems, and corresponding controls that is without CH₄ (Biotic control) or without cells (Abiotic control). Data generated from $n = 3$ biologically independent samples for each group and error bars indicate standard deviation of the mean. **c** The consumption of fatty acids by *Bacillus* sp. WH-2 in the supernatant of *Methylomonas* sp. WH-1 as the electron donor during the 20-day iron reduction process. Data generated from $n = 3$ biologically independent samples for each group and error bars indicate standard deviation of the mean. **d** Electrochemical response of *Bacillus* sp. WH-2 to 1 mM acetate, along with respective controls that is without acetate or without cells. PBS means the phosphate buffer, which replace the addition of acetate.

the pathways for synthesis of a cytochrome *c* and a flavin were identified in the genome of *Bacillus* sp. WH-2, which may be contributing to the direct and the indirect extracellular electron transfers, respectively.

The mechanism of the electron transfer at the cathode was further investigated by employing a three-electrode bioelectrochemical system. The current produced by the CO₂-reducing consortia could reach up to $0.31 \pm 0.03$ mA under −0.4 V (vs. Ag/AgCl) (Fig. 6a). The consumption of bicarbonate was $2.98 \pm 0.19$ mM after 6 days. The setups without bicarbonate (the biotic control) had no obvious current (Fig. 6a), and the dark control also had very low electric current production (Fig. 6a) and bicarbonate consumption ($0.32 \pm 0.10$ mM). According to 16S rRNA gene fragment amplicon sequencing, *Rhodopseudomonas* was the major species in the community on the cathodic biofilm (96%; Supplementary Fig. 11). The 16S rRNA gene sequence of the *Rhodopseudomonas* strain(s) in the CO₂-reducing consortia showed 100% identity with the type strain *Rhodopseudomonas palustris* CGA009, and we employed this strain in our study to confirm the microcosm data. The maximum current produced by *R. palustris* CGA009 was $0.30 \pm 0.04$ mA (Fig. 6b), which was equal to that of the CO₂-reducing consortia. Based on previous reports, *R palustris* CGA009 can obtain electrons directly from solid electrodes[28], including metabolism that couples iron oxidation with CO₂ reduction where light serves as an energy source and Fe(II) as an electron donor[29].

## Discussion

In this study, we demonstrated that the coupled process for both CH₄ oxidation and CO₂ reduction can be realized with the mediation of Fe species and offers substantial benefits in greenhouse gas capture and fixation. While these processes have been observed separately, their joint occurrence was not reported due to energy barriers and lack of electron donors and/or acceptors in most natural environment. With the mediation of Fe species or electrodes, the energy barrier can be overcome, and the CO₂ reduction rate was increased. Furthermore, the regenerable capabilities of Fe species made the electron donors and acceptors readily available and therefore enabled the electrochemical reactions to occur. Additional evidence can be found from metagenomic data from over 40 environmental samples, which revealed the co-existence of key biological groups (*Methylomonas, Bacillus*, and *Rhodopseudomonas*) involved in our study (Supplementary Data 1). As a result, a total of $16.7 \pm 1.2\%$ of carbon was fixed in the biomass, which offers benefits to greenhouse gas mitigation. Our experiments present evidence for a potentially ecologically significant process in which iron minerals function as a nature battery.

While three dominant bacterial species were mainly involved in this process in our specific samples taken from a paddy soil (*Methylomonas, Bacillus* and *Rhodopseudomonas*), additional bacteria with a potential for similar biochemistry are known that are dominant in other types of soils. For example, a consortium of engineered anaerobic methanotrophic archaea (*Methanosarcina acetivorans*) and

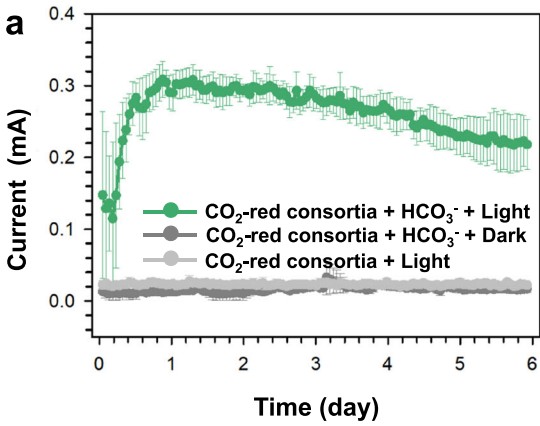

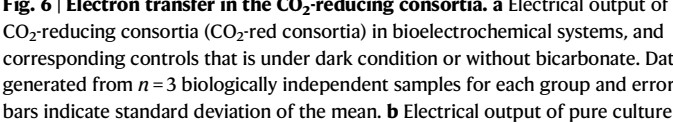

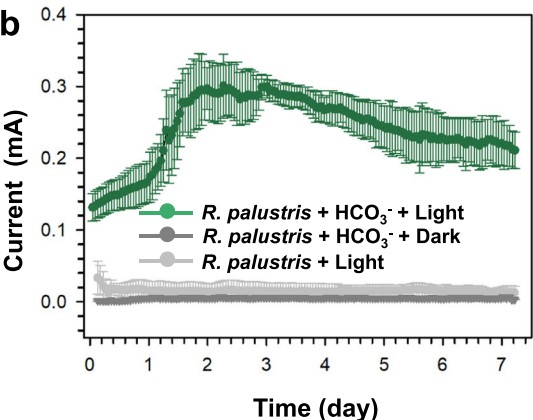

**Fig. 6 | Electron transfer in the $CO_2$-reducing consortia. a** Electrical output of $CO_2$-reducing consortia ($CO_2$-red consortia) in bioelectrochemical systems, and corresponding controls that is under dark condition or without bicarbonate. Data generated from $n = 3$ biologically independent samples for each group and error bars indicate standard deviation of the mean. **b** Electrical output of pure culture *Rhodopseudomonas palustris* CGA009 (*R. palustris*) in the bioelectrochemical system, and corresponding controls that is under dark condition or without bicarbonate. Data generated from $n = 3$ biologically independent samples for each group and error bars indicate standard deviation of the mean.

electroactive bacteria (*Geobacter sulfurreducens*) could convert $CH_4$ into electricity, representing another type of $CH_4$-driven electron-donating consortia[30]. According to reported studies, there are up to 40 known genera of methanotrophs[31] and more than 65 genera of electroactive Fe(III)-reducing bacteria[32–34]. Through cross-feeding, these functional types present a potential for a multitude of combinations of electron-donating consortia driven by $CH_4$. Except for *Rhodopseudomonas*, other microbes such as *Rhodomicrobium*[35], *Rhodobacter*[36], *Chromobacterium*[37] could use Fe(II) minerals as the electron donors and $CO_2$ as the carbon source, which representing another type of $CO_2$-driven electron-accepting consortia. At the same time, not only iron minerals but also other redox substances have a potential to be employed in natural geo-battery to couple $CH_4$ oxidation and $CO_2$ reduction, these processes may play a substantial role in reducing the greenhouse impact.

Further studies are still needed to focus on enhancing the electron transfer efficiency and subsequently increasing the carbon conversion from $CH_4$ and $CO_2$. Firstly, the electrons from $CH_4$ oxidation by the electron-donating consortia can be divided between biosynthesis and extracellular respiration. The biosynthetic process would be beneficial to the $CH_4$ removal, whereas extracellular respiration would help to drive the electron-accepting consortia for $CO_2$ reduction. The energy balances could be further optimized in electron-donating consortia to maximize the benefits of concomitant removal of both $CH_4$ and $CO_2$. Secondly, in order to improve the coulombic efficiency of the anode chamber, it is necessary to further optimize the co-culture of methanotrophs and electroactive bacteria, as acetate serves as an energy carrier between them. Thirdly, the electron transfer on the interface between electron-donating/accepting consortia and iron minerals would determine the coupled efficiency of the two processes. Furthermore, dissolved electron shuttles that are known to aid in long-distance electron transfers[38] may further enhance the electron flow from electron-donating ($CH_4$-oxidizing) to electron-accepting ($CO_2$-reducing) consortia. Overall, iron minerals with excellent regenerable capabilities serves as a recyclable driving force to facilitate the oxidation of $CH_4$ and the reduction of $CO_2$, which could significantly decrease the need for external electron donors or acceptors. At the same time, the iron minerals in this system demonstrate exceptional regenerative properties, allowing them to be reused for driving $CH_4$ oxidation and $CO_2$ reduction. This is a promising approach for establishing a stable and energy-efficient system to eliminate greenhouse gases. Moreover, the phenomenon we described, the previously overlooked mode of $CH_4$ oxidation connected to $CO_2$ reduction in a

mild process. The works presented here proposes a promising strategy for enhancing the efficiency of the sinks for both $CH_4$ and $CO_2$ through interdisciplinary and intelligent engineering designs.

## Methods
### Reagents
Ammonium chloride ($NH_4Cl$), magnesium sulfate heptahydrate ($MgSO_4 \cdot 7H_2O$), calcium chloride dihydrate ($CaCl_2 \cdot 2H_2O$), Sodium phosphate dibasic ($Na_2HPO_4$), potassium dihydrogen phosphate ($KH_2PO_4$), iron (III) chloride hexahydrate ($FeCl_3 \cdot 6H_2O$), sodium hydroxide (NaOH), sodium bicarbonate ($NaHCO_3$), potassium chloride (KCl), potassium hydroxide (KOH), ethanol, Luria-Bertani medium, agarose gel, osmium acid were purchased from Aladdin Industrial Corporation, China. Anhydrophosphoric acid, 4-aminobenzoic acid, hydrochloric acid, glutaraldehyde, acetone, potassium hexacyanoferrate (III) ($K_3[Fe(CN)_6]$), sodium acetate were purchased from Sinopharm Chemical Reagent Co. Ltd. China.

### Soil samples and enrichments
Soil samples were collected from a paddy soil (24° 45′ 22″ N, 118° 4′ 2″ W) using sterile equipment. To enrich for $CH_4$-oxidizing consortia, 2.00 g of soil and 50 mL of ammonium mineral salts (AMS) medium were added to 250 mL serum bottles and atmosphere was created containing 25% ($V/V$) $CH_4$ and 75% ($V/V$) air. The composition of the AMS medium was per liter: $NH_4Cl$ 0.534 g, $MgSO_4 \cdot 7H_2O$ 0.200 g, $CaCl_2 \cdot 2H_2O$ 0.140 g, $Na_2HPO_4$ 0.284 g, $KH_2PO_4$ 0.272 g, 0.2% trace element solution[20]. The cultures were incubated with rotation at 150 rpm at 30 °C. After 3 days, 10% of the culture were transferred into fresh medium and incubated as above for 3 days. The enriched consortia were obtained after 5 transfers. To enrich for $CO_2$-reducing consortia, 2.0 g of soil and 200 mL of modified AMS medium (amended with 10 mM bicarbonate and 3 μM 4-aminobenzoic acid) were added to 250 mL serum bottles and incubated at 30 °C under a 60 W incandescent light source. After the culture turned red (5 days), 10% of the culture were transferred into fresh medium and incubated as above. The enriched consortia were obtained after 5 transfers.

### Construction of communities co-metabolizing $CH_4$ and $CO_2$
**Incubations with soil samples.** A mixture of 0.1 g of the soil sample and 50 mL AMS medium supplemented with 3 μM 4-aminobenzoic acid was placed into 110 mL serum bottles. Ferrihydrite, was prepared by neutralizing 400 mM solution of $FeCl_3$ to a pH of 7 with 5 M NaOH[39], was added to the mixture to a final concentration of 5 mM. The

headspace of serum bottles was filled with the atmosphere of 25% (V/V) $CH_4$, 10% (V/V) air and 65% (V/V) nitrogen. They were incubated with 150 rpm at 30 °C, in the dark. After 14 days, the headspace was pushed with nitrogen to remove $CH_4$ and injected with 15% (V/V) of $CO_2$, and the other condition of culture was not change. The microcosms were incubated at 30 °C, in constant light with a 60 W incandescent light source. The concentrations of $CH_4$ and $CO_2$ were monitored as described below. The soil underwent steam sterilization, serving as the killed control in an incubation with sterilized soil.

**Incubation with the enrichment samples.** The $CH_4$-oxidizing consortia cultures (OD$_{600}$ 0.55) were collected by centrifugation (2655 × g, 5 min), washed three times with AMS medium, and resuspended in AMS medium to OD$_{600}$ of 0.55. The 50 mL cell suspension and 10 mM ferrihydrite were added to 250 mL serum bottles and the atmosphere was adjusted to 25% (V/V) $CH_4$, 5% (V/V) air and 70% (V/V) nitrogen. The cultures were incubated in the dark at 30 °C with 150 rpm. After 8 days of incubation, to exclude the role of extracellular secretion for the next step, cultures were centrifuged (6797 × g, 5 min) in an anaerobic glove box and washed three times with AMS medium. Pellets were resuspended in 50 mL modified AMS medium. 100 mL of the $CO_2$-reducing consortia (OD$_{660}$ 0.70) were centrifuged (2655 × g, 5 min) and washed three times with AMS medium. Cells were added to the mixture described above, and the mixed consortia were incubated at 30 °C in constant light with a 60 W incandescent light source. There were three control experimental setups: the first was prepared in the absence of carbon source ($CH_4$ and $HCO_3^-$) and the second in the absence of consortia during the entire process. The third was in the absence of $CO_2$-reducing consortia after 8 days of dark incubation switched to light incubation. Concentrations of Fe(II) and Fe(total) were monitored, and the structure of iron minerals was analyzed at three time points (0, 8 and 14 days) as described below.

The redox cycles of iron minerals were monitored over several cycles as follows. When the incubation was switched from light to dark, the cultures were centrifuged (6797 × g, 5 min) in an anaerobic glove box, washed three times with AMS medium and resuspended in the appropriate volume of AMS medium for the next cycle. Concentrations of Fe(II) and Fe(total) were monitored every day.

**Chemical measurements.** The concentrations of $CH_4$ and $CO_2$ in the headspace were measured by gas chromatography (FULI GC9790II, China). The following parameters were used for gas chromatography: separation column TDX-01 (2 m × 3 mm); carrier gas argon, 45 mL·min$^{-1}$; column temperature, 120 °C; detector temperature, 160 °C; injection temperature, 160 °C. The total dissolved organic and inorganic carbon of the cultures was measured by total carbon analyzer (TOC-L-CPH, Shmadzu, Japan).

The concentrations of Fe(II) and Fe(total) were measured using the phenanthroline method[20]. For X-ray diffraction, the samples were centrifuged (6797 × g, 5 min) and washed three times with water in a glove box and dried by vacuum freeze-drying. The samples were measured on a Si wafer by X'Pert Pro (PANalytical, Netherlands), equipped with Cu-Kα X-ray tube (40 kV, 40 mA). High-resolution transmission electron microscopy (HTEM) and magnetic measurements of the mineral particles were performed at Palaeomagnetismon Lab of IGG-CAS (Beijing). The HTEM were observed on JEM-2100HR transmission electron microscope (JEM-2100HR, JEOL, Japan). For magnetic measurements, samples were filled into the non-magnetic capsule with a self-made non-magnetic spoon and the measurements were performed using MPMS XP-XL5 (Quantum Design, USA). The hysteresis loop of the sample measured at 5 K with an applied field of 3 **T** and measuring sensitivity of $5.0 \times 10^{-10}$ Am$^2$. The curves of coercivity (Bc), saturation magnetization (Mr) and residual saturation magnetization (Mrs) were obtained after mass normalization.

**Isotopic labeling experiments.** For isotopic labeling, the enriched cultures of $CH_4$-oxidizing consortia (OD$_{600}$ 0.55) were collected by centrifugation (2655 × g, 5 min), washed three times with AMS medium and resuspended in AMS medium to OD$_{600}$ of 0.55. The 50 mL cell suspensions and 10 mM ferrihydrite were added to 250 mL serum bottles, and the headspace was filled with 25% (V/V) $CH_4$ ($^{12}$C), 5% (V/V) air and 70% (V/V) nitrogen. Then 2 mL of $^{13}$C-labeled $CH_4$ (Sigma-Aldrich, 99 atom% $^{13}$C) were injected into the headspace. The cultures were incubated in the dark at 30 °C with 150 rpm. After 8 days, to exclude the role of extracellular secretions for the next step, the cultures were centrifuged (6797 × g, 5 min), washed three times by AMS medium, and resuspended in 50 mL modified AMS medium containing 1 mM $^{13}$C-labeled bicarbonate and 9 mM unlabeled bicarbonate. All the procedures were conducted in an anaerobic glove box. The headspace of the cultures was flushed with nitrogen. One hundred milliliters of the $CO_2$-reducing consortia (OD$_{660}$ 0.70) were centrifuged (2655 × g, 5 min) and washed three times with the modified AMS medium. The cells were mixed with the culture described above and incubated at 30 °C in constant light with a 60 W incandescent light source. The cultures were in absence of cells as the abiotic control and in absence of carbon source ($CH_4$ and $HCO_3^-$) as the biotic control, and all other conditions were identical to those in the experimental setups. The concentration of $CH_4$ and $CO_2$ in gas, dissolved organic/inorganic carbon in liquid, and carbon in biomass of before and after the reaction was measured as follow methods, and the carbon flow percentage was calculated as the ratio of carbon change in each component to original concentration of carbon source ($CH_4$ or $HCO_3^-$).

**Labeled carbon measurements in gas.** The labeled $^{13}CH_4$ and $^{13}CO_2$ in the headspace were measured using the Gas Source Isotopic Ratio Mass Spectrometer (IRMS, MAT253 PLUS, Thermo, USA). The following parameters were used for IRMS: chromatographic column RT-Q-BOND (30 m × 0.32 mm, Agilent, USA), carrier gas helium, gas flow rate 1.2 mL·min$^{-1}$. Column temperature was set at 40 °C and held for 6 min. The oxidation furnace temperature was 1000 °C.

**Labeled carbon measurements in liquid.** Supernatants were collected by centrifuging (6797 × g, 5 min), and the labeled $^{13}$C-total organic carbon (TOC) was measured by a TOC analyzer (isoTOC cube, Elementar, Germany) coupled with IRMS (IsoPrime100, Elementar, UK). The following parameters were used: carrier gas helium was at a flow rate of 100 mL·min$^{-1}$; the oxidation and reduction tubes were kept at 850 °C and 600 °C, respectively; the adsorption and analytical temperature of the $CO_2$ adsorption column was 230 °C; the temperature of IRMS detector was set at 40 °C; the trap current was 300 μA.

The labeled $^{13}$C-total inorganic carbon (DIC) was determined following its quantitative conversion to $CO_2$ by acidification and measured by IRMS (Delta V Advantage, Thermo, USA) with a GasBench II Autosampler (CombiPAL, CTC Analytics, Switzerland). A 12 mL Exetainers (Labco, England) was filled with 400 μL anhydrophosphoric acid and helium (99.999%) was pumped into Exetainers using the exhaust needle of the GasBench II automated sampler for 5 min at a flow rate of 100 mL·min$^{-1}$. Then 0.2 mL of the sample was added and spun at 1699 × g for 2 min. A GasBench II autosampler with a quantification loop (100 μL) was used to sample and separate the $CO_2$ by high purity helium entering the chromatographic column PoraPlotQ (30 m × 0.32 mm, Agilent, USA) at 75 °C and detected by IRMS.

**Labeled carbon measurements in biomass.** The samples were centrifugated (6797 × g, 5 min) and the pellets were collected for measuring the particulate organic carbon. The pellet samples were cleaned twice with 1 M HCl to remove residual inorganic carbon and dried with vacuum freezer. The particulate organic carbon of the biomass was analyzed with an elemental analyzer (Flash EA 2000, Thermo, Germany) combined with IRMS (MAT253 Plus, Thermo, USA). The carbon

in biomass was converted to $CO_2$ by an oxidation catalyst and the reduction tube in the elemental analyzer. The oxidation and reduction tubes were kept at 960 °C, and oxygen at a rate of 175 mL·min⁻¹. The $CO_2$ were directed by the carrier gas helium (100 mL·min⁻¹) towards the IRMS.

**Microbial community analysis.** Genomic DNA was extracted using the DNeasy Isolation PowerSoil Kit (Qiagen, Germany) according to manufacturer's instructions. The hypervariable regions V3-V4 of the bacterial 16S rRNA gene were amplified with the primer pair 338F (5′-ACTCCTACGGGAGGCAGCAG-3′) and 806R (5′-GGACTACHVGGGTW TCTAAT-3′). The PCR product was extracted from 2% agarose gel, purified using the AxyPrep DNA Gel Extraction Kit (Axygen Biosciences, Union City, CA, USA) and quantified using Quantus™ Fluorometer (Promega, USA). Purified products were pooled in equimolar amounts, and paired-ended sequencing was performed on an Illumina MiSeq PE300 platform (Illumina, San Diego, USA) according to the standard protocols by Majorbio Bio-Pharm Technology Co. Ltd. (Shanghai, China). The amplicon sequences were analyzed with QIIME 2 (v2023.2)[40]. The low-quality reads (less than 100 bp or quality score less than 25) were removed by Trimmomatic (v0.39). Amplicon sequence variants (ASVs) were clustered applying DATA2. The representative sequences of ASVs were matched against the SILVA database (https://www.arb-silva.de) for taxonomic assignments. The microbial composition was resolved at the genus level.

**Isolation and cultivation of pure cultures.** As *Methylomonas* and *Bacillus* dominant in the biofilm of the anode, they were isolated as pure cultures using a gradient dilution method. For obtaining pure *Methylomonas*, $CH_4$-oxidizing consortia were spread onto solid AMS medium (1.5% agar) and incubated in an air-tight tank under the atmosphere of 25% (V/V) $CH_4$ and 75% (V/V) air at 30 °C. For obtaining pure *Bacillus*, the consortia were spread onto solid Luria-Bertani medium (1.5% agar) and incubated at 30 °C. A single colony was transferred to fresh medium, then diluted 100 and 10000 times before being dispersed onto new solid medium. The dilutions were repeated until obtaining a pure colony. The two strains were named, respectively, *Methylomonas* sp. WH-1 and *Bacillus* sp. WH-2. Their purity was checked by microscopy (Olympus CX23, USA) and by 16S rRNA sequencing. Since *Rhodopseudomonas* was dominant in the biofilm of the cathode, stain *Rhodopseudomonas palustris* CGA009 with a highly similar 16S rRNA (100%)[41] was employed, obtained from Prof. Yanning Zheng (IM-CAS, Beijing).

The cells or biofilm for scan electron microscopy (SEM) analysis were fixed in 2.5% (W/V) glutaraldehyde phosphate buffered solution (100 mM, pH 7.0) for 24 h at 4 °C. Then the samples were washed three times with phosphate buffered (100 mM, pH 7.0). Next, the cells were dehydrated in a gradient ethanol series (0% twice, and 30, 50, 70, 90, 95, and 100% twice) for 15 min each. After drying with critical point drying for 12 h, the cells were placed onto a carbon substrate for SEM observation (S-4800 FE-SEM, Hitachi, Japan). For observation of the transmission electron microscopy (TEM), the samples were treated with 2.5% glutaraldehyde (4 h, 4 °C), 1% osmium acid (1 h, 4 °C), and ethanol and acetone, respectively[42]. Then samples were embedded in Supur resin and cut in an ultramicrotome (Leica UC7, Germany), and observed using a TEM (H-7650, Hitachi, Japan).

**Genomic analysis of pure cultures.** Genomic DNA of the pure cultures was extracted using TIANamp Bacteria DNA Kit (TIANGEN, China). The quality of DNA was evaluated by Nanodrop (ND-1000) Spectrophotometer and by gel electrophoresis. PE150 DNA library was constructed for Illumina sequencing (Majorbio, China). Raw data were trimmed and filtered by NGSQCToolkit (v2.3). Replicated reads were removed by FastUniq (v1.1). High-quality reads were further corrected by BLESS (v1.01). Edena (v3.131028) was used for genome assembly.

Assembled scaffolds were annotated by Prokaryotic Genome Annotation Pipeline platform (PGAP, NCBI).

## Construction of bioelectrochemical systems (BES)

**BES connecting $CH_4$-oxidizing consortia and $CO_2$-reducing consortia.** This BES had two chambers, including the anodic chamber and the cathodic chamber, separated by a proton exchange membrane (Zhejiang Qianqiu Water Treatment, China). Both the anode and the cathode were carbon brush (3 cm diameter, 3 cm height, Hubei HOT-Material, China), and the anolyte was the AMS medium and the catholyte was the modified AMS medium. The headspace of the anode was filled with 25% $CH_4$ (V/V, 75% air) as the electron donor, and 10 mM $NaHCO_3$ was added into the catholyte as the electron acceptor. The $CH_4$-oxidizing consortia ($OD_{600}$ 0.55) and the $CO_2$-reducing consortia ($OD_{660}$ 0.70) were collected by centrifugation (2655 × g, 5 min) and washed three times with AMS medium. Then $CH_4$-oxidizing consortia and the $CO_2$-reducing consortia were inoculated into the 110 mL anolyte and the 110 mL catholyte, respectively. A stirring bar was placed at the anode chamber incubated at 30 °C and stirred at 200 rpm on using a magnetic stirrer, to aid in dissolution of the $CH_4$ gas, and the chamber was covered with tin foil to maintain darkness. The BES was incubated at 30 °C under a 60 W light source. The external load of BES was 500 Ω resistance and the output voltage of BES was recorded every 5 min by a digital multimeter (Keithley Instruments, USA). There were two control experimental setups: the one was prepared in the absence of $CH_4$ in the anode and $HCO_3^-$ in the cathode (biotic control), and the another in the absence of cells in the two chambers (abiotic control). The all-other conditions were the same as above. The Polarization curves of BES were obtained by replacing the external resistance method when the voltage of BES ran stably. The external resistance was 10000, 5100, 2000, 1500, 1000, 800, 620 Ω, respectively, in turn from large to small change the resistance with 30 min of interval time and collected voltage every 1 min. Power was calculated by $P = VI$, where $V$ was the voltage of BES, and $I$ was the voltage apart from resistance.

**BES of $CH_4$-oxidizing consortia.** This BES was constructed as described above for the two-chamber BES, except for the mixture of 50 mM $K_3[Fe(CN)_6]$ and AMS medium was used as a catholyte solution.

Cells of *Methylomonas* sp. WH-1 ($OD_{600}$ 0.55) and *Bacillus* sp. WH-2 ($OD_{600}$ 1.80) were collected by centrifuging (2655 × g, 5 min) and washed three times with AMS medium, respectively. Then the cells of two strains were mixed and resuspended in 110 mL anolyte. As the control of a single culture, cells of either *Methylomonas* sp. WH-1 or *Bacillus* sp. WH-2 were inoculated in the anolyte, respectively. The headspace of the anode was filled with 25% $CH_4$ (V/V) and 75% air (V/V), and the cathode was filled with air. A stirring bar was in the anode stirring at 200 rpm on a magnetic stirrer and the BES was cultivated at 30 °C. The external load of BES was 500 Ω resistance and the output voltage was recorded every 5 min by a digital multimeter (Keithley Instruments, USA). There were two control experimental setups: the one was prepared in the absence of $CH_4$ (biotic control) and the another in the absence of cells (abiotic control). The all-other conditions were the same as above.

**BES of $CO_2$-reducing consortia.** This BES was a three-electrode system, in which the working electrode was carbon brush (3 cm diameter, 3 cm height), the counter electrode was carbon felts (3 × 5 × 0.5 cm, Haoshi Carbon Fiber, China), and the reference electrode was Ag/AgCl electrode (KCl saturated). The electrolyte was the modified AMS medium. Cells of the $CO_2$-reducing consortia ($OD_{660}$ 0.70) were collected by centrifugation (2655 × g, 5 min), washed three times with the modified AMS medium, and resuspended in 200 mL of the electrolyte incubated at 30 °C. The working electrode was the cathode, which was poised at a potential of −0.4 V (vs. Ag/AgCl), and the current was recorded by an electrochemical workstation (CHI1000C, Chenhua,

China). The BES was placed in the dark as a control setup and prepared in the absence of bicarbonate as a no-electron acceptor control, with all other conditions remaining the same as described above.

**Metabolomics.** For understanding the cross-feeding interaction between *Methylomonas* and *Bacillus*, the supernatant of *Methylomonas* sp. WH-1 ($OD_{600}$ 0.55) was separated from cells by centrifugation ($3824 \times g$, 10 min), followed by filtration (0.22 μm). Cells of *Bacillus* sp. WH-2 were centrifuged ($2655 \times g$, 5 min), washed three times by phosphate buffer solution, and resuspended in supernatant of *Methylomonas* sp. WH-1 to $OD_{600}$ 1.8. The 10 mM ferrihydrite was added into the culture, and serum bottles were sealed with butyl rubber stoppers. The cultures were flushed with nitrogen for 5 min to remove oxygen and incubated at 30 °C at 150 rpm. 2 mL samples of the cultures were taken at days 0 and 20. These were centrifuged ($6797 \times g$, 10 min) to collect the supernatant. The supernatant was filtered through 0.22 μm filters and acidified to pH lower than 2 by adding 1 M HCl for measuring volatile fatty acids (formate, acetate, lactate, propionate, butyrate, isobutyrate, valerate, isovalerate). The concentrations of the volatile fatty acids were measured by ion chromatography (ICS-3000, Dionex, USA) equipped with a guard column (4 mm × 50 mm) and an anion exchange column (Dionex IonPacTM AS11-HC, 4 mm × 250 mm), at 30 °C. The mobile phase was milli-Q water with the $1.0 \text{ mL·min}^{-1}$ of flow rate and the leachate was 1 mM KOH.

**Chronoamperometry.** The response of *Bacillus* sp. WH-2 to acetate was recorded by chronoamperometry on electrochemical workstation (CHI 832, Chenhua, China). The three electrodes system of chronoamperometry consisted of a glassy carbon electrode (5 mm in diameter) as working electrode, an Ag/AgCl (saturated KCl) as reference electrode and a platinum wire as counter electrode. The cells of *Bacillus* sp. WH-2 were centrifuged ($2655 \times g$, 5 min) and washed three times by phosphate buffer. Then, 5 μL of cell suspension was dropped onto the working electrode. The working electrode was applied with the potential of 0.3 V (vs Ag/AgCl). When the current reach to stable, 200 μL of 1 M sodium acetate was added into the electrolyte (phosphate buffer solution, $3.4 \text{ g·L}^{-1}$ $KH_2PO_4$ and $3.55 \text{ g·L}^{-1}$ $Na_2HPO_4$). For a no-electron donor control, 200 μL of electrolyte was added into the electrolyte. For an abiotic control, no cells were dropped onto the working electrode.

**Statistics and reproducibility**

All experiments were replicated a minimum of three times independently, with consistent results. The figure legends indicate the number of biologically independent samples (*n*). The data were subjected to statistical analysis using a two-tailed Student's *t*-test. A difference was considered statistically significant when the *P*-value was less than 0.05 or less than 0.01. No statistical method was used to predetermine sample size. No data were excluded from the analyses. The experiments were not randomized. The Investigators were not blinded to allocation during experiments and outcome assessment.

**Reporting summary**

Further information on research design is available in the Nature Portfolio Reporting Summary linked to this article.

## Data availability

The iTag sequencing data generated in this study have been deposited in the NCBI Sequence Read Archive (SRA) database under the BioProject accession code PRJNA1087527 and corresponding BioSample accession codes SAMN40446864-SAMN40446878. The genome of *Methylomonas* sp. WH-1 used in this study is available in the NCBI Genome database under BioProject accession code PRJNA705869 and BioSample accession code SAMN18105290. The genome of *Bacillus* sp. WH-2 used in this study is available in the NCBI Genome database

under BioProject accession number PRJNA705873 and BioSample accession number SAMN18105338. Source data are provided with this paper.

## Code availability

The custom scripts used for analyzing data are available at GitHub: https://github.com/yuezheng90/CH4CO2-project/tree/main.

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

## Acknowledgements

This work was supported by the National Natural Science Foundation of China (22025603 to F.Z., 42021005 to F.Z., 41907027 to Y.Z., 41890843 to J.H.L. and 91851102 to Y.N.Z.) and Nanqiang Young Talents Supporting Program (Xiamen university, Y.Z.).

## Author contributions

Conceptualization: Y.Z. and F.Z. Methodology: Y.Z., H.W., Y.L., P.Y.L., B.L.Z., Y.N.Z. and J.H.L. Visualization: Y.Z. and H.W. Funding acquisition: Y.Z., F.Z., J.H.L. and Y.N.Z. Project administration: Z.J.R. and F.Z. Writing-original draft: Y.Z., H.W. and F.Z. Writing-review and editing: Y.Z., L.C., Z.J.R. and F.Z.

## Competing interests

The authors declare no competing interests.
