## [Peer Review File · Nature Communications]

REVIEWERS' COMMENTS

Reviewer #1 (Remarks to the Author):

My comments have been well addressed in the revised manuscript, so no further comments from my side.

Reviewer #2 (Remarks to the Author):

Reviewer thinks that authors appropriately revised manuscript with considering reviewers' concern and suggestions.

Reviewer has just one thing to suggest is:

Regarding Question 7 of Reviewer1, reviewer is asking about the short time period of the system operation (~ 0.5 day), and current decrease in the time. If the suggested mechanism is working and sustainable in the realistic system, the time should be reasonably longer?

Manuscript ID: NCOMMS-24-08871-T

Title: Electrochemically coupled CH₄ and CO₂ consumption driven by microbial processes

Response to Reviews

Reviewer #1 (Remarks to the Author):

My comments have been well addressed in the revised manuscript, so no further comments from my side.

Response:

We are grateful to the support from the reviewer and appreciate the reviewer's recommendation.

Reviewer #2 (Remarks to the Author):

Reviewer thinks that authors appropriately revised manuscript with considering reviewers' concern and suggestions.

Reviewer has just one thing to suggest is:

Regarding Question 7 of Reviewer1, reviewer is asking about the short time period of the system operation (~ 0.5 day), and current decrease in the time. If the suggested mechanism is working and sustainable in the realistic system, the time should be reasonably longer?

Response:

We appreciate the positive feedback. For the remaining question on the duration, we observed that methane and bicarbonate, as the electron donor/acceptor, respectively, both are still available at the end of the BES cycle, as shown in Figure S6. The failure of the methane-oxidizing communities in the anode chamber to supply the anode with adequate electrons should be the cause of the current drop. Methanotrophs do not carry electrons directly to the anode in the anodic communities. Acetate, a crucial bridge in anode electron transport, is secreted by methanotrophs to *Bacillus* as an energy carrier (Figure 5). We think that the drop in acetate concentration is what's causing the current to decrease. The section on discussion has been expanded to highlight the significance of optimizing the co-culture between *Methylomonas* species and *Bacillus* species in the future.

Line 272: Secondly, in order to improve the coulombic efficiency of the anode chamber, it is necessary to further optimize the co-culture of methanotrophs and electroactive bacteria, as acetate serves as an energy carrier between them.